# VOXELKP: A VOXEL-BASED NETWORK ARCHITECTURE FOR HUMAN KEYPOINT ESTIMATION IN LIDAR DATA

## ABSTRACT

We present *VoxelKP*, a novel fully sparse network architecture tailored for human keypoint estimation in LiDAR data. The key challenge is that objects are distributed sparsely in 3D space, while human keypoint detection requires detailed local information wherever humans are present. First, we introduce a dual-branch *fully sparse spatial-context block* where the spatial branch focuses on learning the local spatial correlations between keypoints within each human instance, while the context branch aims to retain the global spatial information. Second, we use a *spatially aware multi-scale BEV fusion* technique to leverage absolute 3D coordinates when projecting 3D voxels to a 2D grid encoding a bird's eye view for better preservation of the global context of each human instance. We evaluate our method on the Waymo dataset and achieve an improvement of 27% on the MPJPE metric compared to the state-of-the-art, *HUM3DIL*, trained on the same data, and 12% against the state-of-the-art, *GC-KPL*, pretrained on a $25\times$ larger dataset. To the best of our knowledge, *VoxelKP* is the first single-staged, fully sparse network that is specifically designed for addressing the challenging task of 3D keypoint estimation from LiDAR data, achieving state-of-the-art performance. Our code is available at `https://`.

## 1 INTRODUCTION

Human pose estimation is a critical area of research with applications spanning computer vision, robotics, human-computer interaction, and augmented/virtual reality. Previous works (Toshev & Szegedy, 2014; Newell et al., 2016; Sun et al., 2019) are mostly based on 2D images and videos. Compared to regular RGB input, LiDAR sensors provide detailed 3D structural information by measuring the distance to objects using laser light. Apart from its robustness under occlusion and illumination changes, LiDAR also offers privacy protection as it can not retain facial details. In recent years, significant progress has been made in 3D object detection from LiDAR point clouds, with methods like PointRCNN (Shi et al., 2019a), Part-A2 (Shi et al., 2019b), and PV-RCNN (Shi et al., 2020) achieving impressive results, while human pose estimation from LiDAR is still an open research problem with much room for improvement. Typically, object detection methods focus on capturing objects scattered sparsely across the 3D space while the keypoints tend to be distributed densely within localized regions around the human body. This fundamental discrepancy in the context captured by existing detectors limits their suitability for precise 3D keypoint prediction due to the lack of fine-grained spatial information. To address this gap, we aim to extend the success of 3D object detection to 3D keypoint estimation for Lidar point cloud data by introducing novel components to preserve fine-grained spatial information. As shown in Figure 1, our method significantly improves the precision of the estimated keypoints.

This work identifies the importance of learning from spatial information of varying densities to capture the intricate spatial relationships between keypoints for precise human pose estimation. For this purpose, we introduce the *VoxelKP* architecture. *VoxelKP* is a novel, fully sparse neural network tailored specifically for human keypoint estimation within LiDAR point clouds. We first introduce a dual-branch *fully sparse spatial-context block* that integrates local dense features with the global spatial context from the sparse representations of LiDAR scans. More precisely, the spatial branch is used to extract local spatial correlation between keypoints whilst the context branch is used to

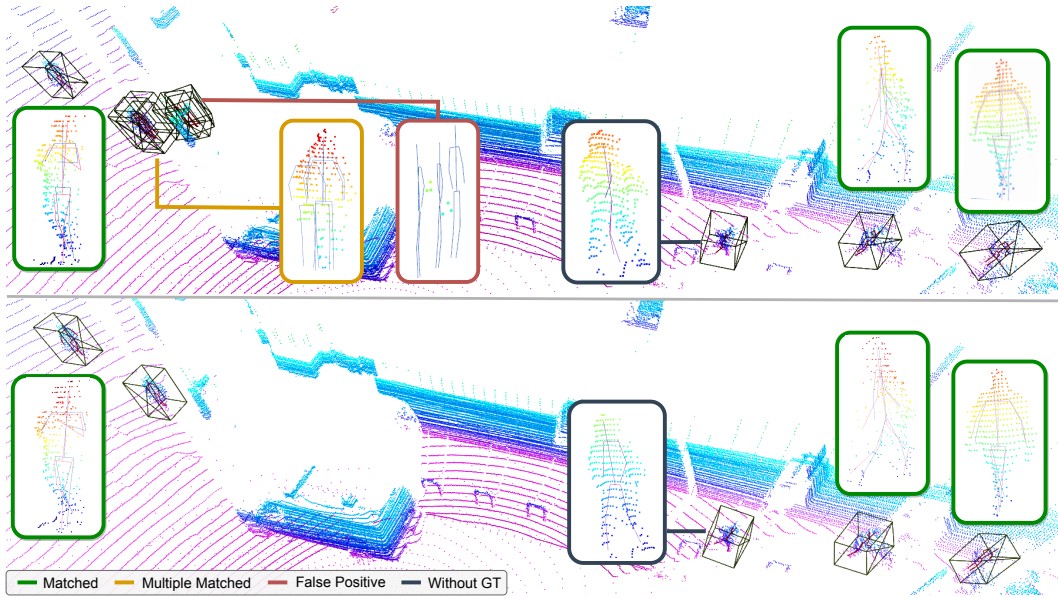

| — Matched | — Multiple Matched | — False Positive | — Without GT |

Figure 1: A visual demonstration of our baseline model (top) and the proposed *VoxelKP* (bottom). Our *VoxelKP* offers improved keypoint estimation with precise locations and fewer false positives. The insets are color-coded according to the legend in the figure. In the green-colored insets, a comparison with the ground truth is shown, with ground truth in red and predictions in blue. Our baseline model is *VoxelNeXt* with additional keypoint estimation outputs.

preserve the global spatial details. Second, we propose a *spatially aware multi-scale BEV fusion* module that aims to effectively encode absolute 3D coordinates to BEV representations, to be better aware of the 3D spatial relationship within 2D BEVs. To the best of our knowledge, *VoxelKP* is the first single-staged, fully sparse network that is specifically designed for addressing the challenging task of 3D keypoint estimation from LiDAR data, achieving 27% on the MPJPE metric compared to the current state-of-the-art trained on the same data.

## 2 RELATED WORK

### 2.1 DEEP LEARNING ON POINT CLOUDS

Many neural network architectures have been adapted for processing point clouds. Earlier methods like VoxNet (Maturana & Scherer, 2015) applied 3D CNNs to voxel grids for object classification. PointNet (Qi et al., 2017a) was one of the first works to operate directly on point clouds using MLPs and max pooling to extract global features of entire scenes represented by point clouds. Follow-up works like PointNet++ (Qi et al., 2017b) introduced hierarchical and localized feature learning. Meanwhile, another branch of works such as PointCNN (Li et al., 2018) and KPConv (Thomas et al., 2019) introduced novel convolutional operators for learning features on the unordered point clouds, overcoming the limitations of typical convolutions for this irregular data type.

Typical LiDAR-generated point clouds contain more than 100,000 points, making point-by-point computations overwhelming due to the massive data scale. VoxelNet (Zhou & Tuzel, 2018) proposed a voxel feature encoding (VFE) layer as a workaround for the high computational and memory issues brought by point-by-point computations. Meanwhile, sparse and submanifold sparse convolution operations (Graham et al., 2018) exploit sparsity in the voxel grid to reduce computations. SECOND (Yan et al., 2018) introduced an efficient sparse convolutional approach that benefits from the sparse operations. Following SECOND, subsequent works like PointPillars (Lang et al., 2019), 3DSSD (Yang et al., 2020), PV-RCNN Shi et al. (2020), CenterPoint (Yin et al., 2021) further advanced sparse convolutional detection on point clouds, introducing ideas like pillar encoding for faster detection, multi-scale detection stacks with anchor boxes, shared voxel encoders, and detecting small objects by center points. VoxelNeXt (Chen et al., 2023) further demonstrates a fully sparse

voxel-based method without sparse-to-dense conversion or NMS post-processing. However, these approaches are targeted at improving bounding box localization accuracy, which does not require fine-grained spatial features for precise keypoint estimation tasks. Instead, We propose *VoxelKP*, a novel sparse convolutional architecture tailored for learning discriminative local features from sparse LiDAR data for accurate human pose estimation.

## 2.2 HUMAN POSE ESTIMATION ON POINT CLOUDS

Human pose estimation has been extensively studied in images, with methods like DeepPose (Toshev & Szegedy, 2014), Stacked Hourglass (Newell et al., 2016), and HRNet (Sun et al., 2019) achieving high accuracy on benchmarks like COCO-wholebody (Jin et al., 2020). However, compared to RGB images, point clouds provide explicit 3D structural information about the shape and depth of objects. Shotton et al. (2011) pioneered point cloud human pose estimation from a single depth image. Recent works such as Zhou et al. (2020); Ma et al. (2021) proposed a deep learning-based 3D human pose estimation from depth images. Waymo (Sun et al., 2020) has released keypoint annotations for LiDAR-collected point cloud scenes, while only $3\%$ of the frames are annotated with keypoint human poses. Due to the scarcity of the keypoint annotations within LiDAR point cloud data, many works have taken semi-supervised or weak-supervised approaches to compensate for the limited availability of labeled 3D pose data. For example, Zanfir et al. (2023); Zheng et al. (2022) took a multi-modal approach to utilize the enriched image annotations to assist the recognition from point clouds. Weng et al. (2023) proposed an unsupervised approach that generates pseudo ground truth without using annotated keypoint data, along with a fine-tuning approach that pretrains the model with synthetic data and then fine-tunes on the training set. A concurrent work (Ye et al., 2023) adopted a fine-tuning strategy that used a frozen backbone pretrained on a large-scale dataset as a feature extractor, achieving plausible performance. In general, multi-person pose estimation from sorely point clouds remains relatively unexplored due to the lack of ground-truth 3D human pose annotations. This work proposes a single-staged keypoint estimation method with only LiDAR point clouds, achieving comparable performance without extra training data.

## 3 METHOD

LiDAR point clouds typically contain sparsely distributed objects that occupy only small regions of the full 3D space. While the distribution of humans in space is sparse, in contrast, human keypoints require dense information wherever a human is present. To handle this density variation, we aim to improve the learning of spatial details in the regions where keypoints need to be located and detailed information is required. As illustrated in Figure 2, our *VoxelKP* framework contains two key components: 1) fully sparse spatial-context blocks, and 2) spatially aware multi-scale BEV fusion. In this section, we first present the formulation of the task, then introduce the key components proposed in our network, and finally elaborate on the details of the network architecture.

## 3.1 PROBLEM FORMULATION

Given a 3D point cloud scanned by LiDAR sensors, our goal is to estimate the 3D locations of $K$ keypoints that represent the human pose. Let the input point cloud $P$ be $\mathbb{R}^{N \times C}$ where $N$ is the number of points and $C$ is the number of features (*e.g.* x, y, z, intensity, elongation). We use a sparse voxel representation to represent point clouds, which consists of two separate tensors: one feature tensor $\mathbb{R}^{V \times C}$ and one index tensor $\mathbb{R}^{V \times 4}$ where $V$ is the number of non-empty voxels and 4 dimensions are used for batch sample index and the three coordinates of each voxel. We define the ground truth pose for the $i^{th}$ human as a set of 3D keypoint locations $G_i = \{g_i^1, g_i^2, ..., g_i^K\}$ where $g_i^k \in \mathbb{R}^3$ is the location of the $k^{th}$ keypoint in the global coordinate frame. The set of $K$ keypoints corresponds to anatomical joints of interest such as shoulders, elbows, wrists, hips, knees, and ankles. Our objective is to predict the 3D keypoint locations from the input point cloud, i.e. to learn a function F such that $\hat{G} = F(P)$, where $\hat{G} \in \mathbb{R}^{M \times K \times 3}$ is the tensor of predicted 3D keypoint locations of $M$ humans.

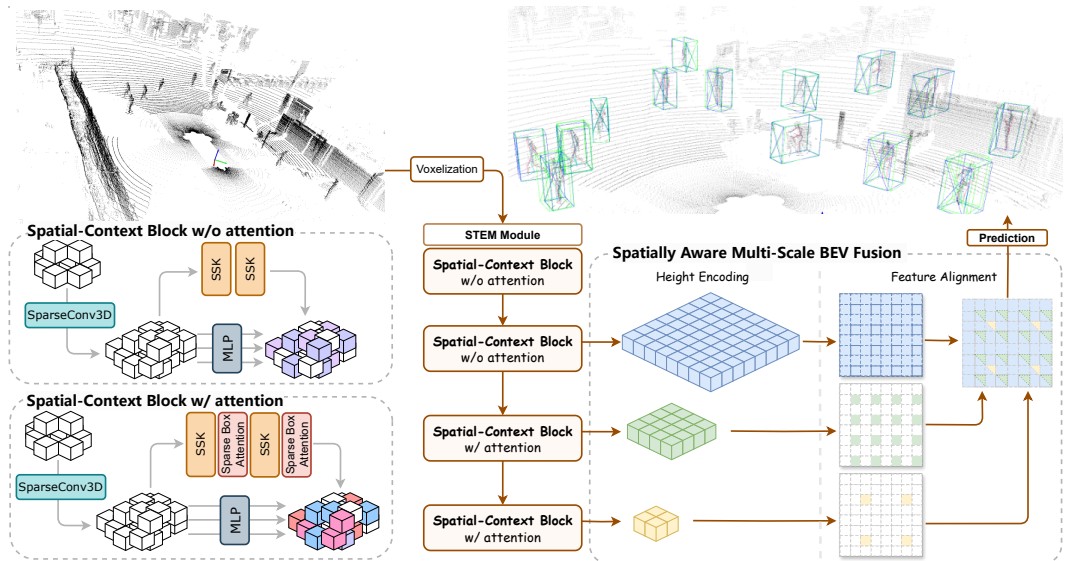

Figure 2: The overall architecture of *VoxelKP*. The model begins with voxelizing a point cloud scene, followed by feature extraction using a stem module (Appendix A.1). Subsequently, the extracted features are processed through four fully sparse spatial-context blocks (Section 3.2) for capturing local and global spatial information. Lastly, we utilize a spatially-aware multi-scale BEV representation (Section 3.3) for accurate human keypoint estimation.

## 3.2 FULLY SPARSE SPATIAL-CONTEXT BLOCK

Our approach is designed with specialized building blocks to process spatial information of varying densities effectively. Each building block starts with a basic sparse 3D block, subsequently branching into two distinct pathways for local and global spatial feature learning. Spatial branches are used for learning local spatial correlations between keypoints, incorporating *sparse selective kernel modules* and *sparse box-attention modules* to improve the representational power to encode and localize the intricate keypoint features. Meanwhile, global spatial feature learning is achieved through context branches, where a straightforward MLP is employed to maintain detailed per-voxel information, ensuring the retention of spatial details. Our proposed *hybrid feature learning* strategy captures the nuanced local details with an understanding of the global context.

### 3.2.1 SPARSE SELECTIVE KERNEL MODULE

Inspired by Li et al. (2019), we propose the sparse selective kernel (SSK) module that selectively aggregates multi-scale features to improve spatial context. By selectively combining semantic information from different scales, the network learns better local dense features of the spatial locations of keypoints. The SSK modules perform spatial attention on a 3D sparse voxel space, where the attention specializes the receptive field at each position using a data-driven kernel selection. As demonstrated in Fig. 3, we first use two sparse 3D submanifold convolution branches with varied receptive field sizes of $3\times3\times3$ and $5\times5\times5$. A submanifold convolution computes output values only if the convolution kernel

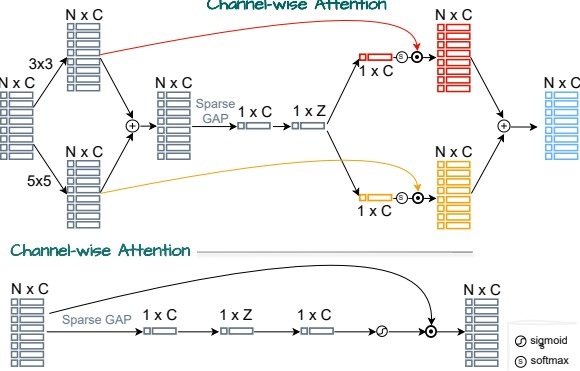

Figure 3: **Sparse selective kernel** module with one sample input. The SSK module selects the best kernels from different receptive fields with a softmax-based channel-wise attention mechanism.

is centered on a non-empty voxel, i.e., the number of non-empty voxels remains the same. These operations are applied to sparsely sampled voxel locations, extracting multi-scale features while remaining efficient. Next, the features from each branch are summed up and then fed into a selection module that compresses the spatial dimension by a sparse global average pooling (GAP) to compute the average feature of all non-empty voxels, then a feature squeeze and expansion are applied (Hu et al., 2018). The squeeze and expansion process compresses a feature map from $C$ channels to $Z$ channels, then expands it back to $C$ channels, where $Z$ is 25% of $C$ in our implementation. This produces channel-wise attention weights after a softmax activation, allowing the network to emphasize or suppress the features from each branch selectively. In the end, the multi-scale local features can then be obtained by combining the weighted features from all branches through averaging.

### 3.2.2 Sparse Box-Attention Module

To better capture local dense features, we apply box-based self-attention. Intuitively, the keypoint location is only relevant to its local surrounding regions, of which the global context can barely help. In fact, as demonstrated in Section 5.2, the integration of global features can even harm the estimation accuracy. Thus, unlike the previous works that tried to capture a wider range of global features with self-attention methods for segmentation tasks (Lai et al., 2022; 2023), we focus on localized feature attention to resolve the densely distributed keypoints in local regions.

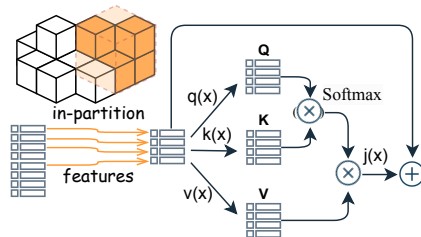

Figure 4: **Sparse box-attention**. This attention mechanism selects the voxel features that correspond to one box partition referring to the index tensor and then performs self-attention on the selected voxels. The functions $q$, $k$, $v$, and $j$ are linear layers.

The key idea is to partition the sparse 3D voxel space into non-overlapping boxes. Within each local box, we apply self-attention to capture dependencies between the voxels inside the box. The features in each box go through a linear layer for the queries $Q$, keys $K$, and values $V$, where $Q, K, V \in \mathbb{R}^{n_b \times h \times d}$ and $n_b, h, d$ are the number of valid voxels in the $b$-th box, attention heads, and feature dimensions. Since we are using sparse tensor representations, each box partition may contain a varying number of voxels. Referring to Lai et al. (2022); Zhang et al. (2022), we then compute the attention map by the following equation:

$$\text{attn}_{i,j,h} = Q_{i,h} \cdot K_{j,h}, \quad \hat{\text{attn}}_{i,.,h} = \textit{softmax}(\text{attn}_{i,.,h}), \quad y_{i,h} = \sum_{j=1}^{n_b} \hat{\text{attn}}_{i,j,h} \times V_{j,h}. \quad (1)$$

The output is then obtained by applying a projection layer and a residual connection, as shown in Fig. 4.

### 3.2.3 Hybrid Feature Learning

The convolutional operations focus on understanding spatial hierarchies and local geometric structures to extract local neighborhood information. Concurrently, inspired by the previous point-voxel networks (Liu et al., 2019; Shi et al., 2020; Zhang et al., 2022), we include an MLP branch for each stage. The integration of an MLP branch alongside a convolutional branch is a strategic approach to capture both fine-grained per-voxel details and relatively coarse-grained local neighborhood information. Each MLP branch is composed of three sequential blocks, each consisting of a linear layer, batch normalization, and a ReLU activation function. The number of channels in each linear layer is set to match the channels of the incoming tensor. We then merge the output features from the MLP and convolutional branches through element-wise summation to create hybrid features of the per-voxel and per-neighborhood information. This hybrid feature learning approach is deployed to retain and process fine details across the voxel space, which is critical for the accurate localization of keypoints.

### 3.3 Spatially Aware Multi-Scale BEV Fusion

Compressing features into bird's eye view (BEV) maps is a common practice for object detection Chen et al. (2017); Yan et al. (2018) to collapse the point cloud to 2D for efficiency. For a

sparse 3D voxel grid of size $C \times X \times Y \times Z$, we use $C$ to denote the number of features per voxel, $X$ and $Y$ as the spatial extent in the ground plane, and $Z$ as the up axis. Starting with a sparse 3D voxel grid, previous works such as Chen et al. (2023) simply ignore the height information by summing the features of all voxels that share the same position on the ground plane (the same $x$ and $y$ coordinates). As shown in Table 2, by employing the spatially aware fusion technique, the performance can be significantly improved over the naive fusion approach. However, different from object detection tasks, height information is essential for keypoint estimation tasks to precisely locate each keypoint. A reasonable approach is to directly deploy 3D feature maps. Unfortunately, this direct 3D approach does not lead to a decent performance as training does not converge well, as shown in Table 3. We, therefore, propose a *spatially aware multi-scale BEV fusion* approach for fusing features from multiple encoder layers in a way that retains spatial information, as illustrated in Fig. 5. Specifically, we use height encoding and scale-wise feature alignment to compensate for the loss of spatial information during the 2D projection.

***Height Encoding*** Transforming 3D data into BEV is often used in 3D object detection and segmentation tasks, for reducing the dimensionality of point clouds and making them more manageable for processing. An object detection method may project the 3D voxel grid to a 2D BEV representation by adding features from voxels that share the same x and y position, losing the information about which height a feature was taken from. Instead, we use a height encoding method. Specifically, we compress the height dimension to 1 using convolution kernels of size $(1, 1, h)$ where $h$ is the height of each 3D voxel grid. Meanwhile, we increase the number of resulting channels to retain more spatial details and features from the 3D representation. This provides a richer representation for the 2D regression heads to work with.

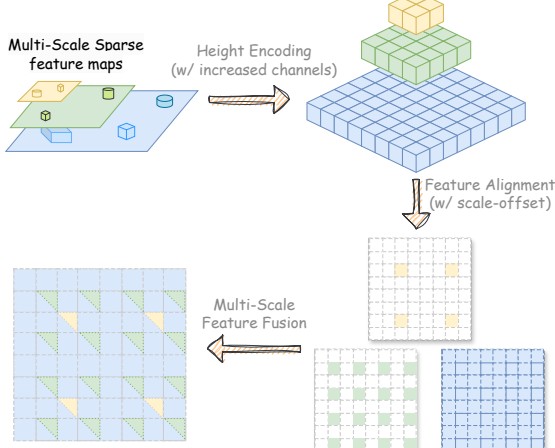

Figure 5: **Spatially aware multi-scale BEV Fusion**. Note that we use a dense representation for a better visual illustration of the method.

***Multi-scale Feature Alignment*** After obtaining $z$ multi-scale height-encoded BEV maps from the last few stages of the network, we then fuse those feature maps to create a feature map that contains multi-scale features. Unlike working with dense tensors, the direct interpolation of the feature maps in the sparse case is computationally complex, as it requires specialized algorithms to efficiently navigate through the predominantly empty voxels to find and interpolate the adjacent non-empty voxels. Instead, we directly modify the feature position of the sparse tensor by multiplying the voxel position by its scale $r$. To avoid overlapping feature positions of $(x_p * 2^r, y_p * 2^r)$ $r \in \{0, 1, 2...\}$ during the scale multiplication, we align the xy-plane positions $(x_p, y_p)$ using scale offsets $(x_p * 2^r + r, y_p * 2^r + r)$, where $p$ is the position of a voxel in a voxel grid.

By stacking the $r$-scaled feature maps together, we obtain a multi-scale 3D feature map with a height of $r$. To obtain a BEV feature map, instead of collapsing with $1 \times 1 \times r$ convolutions, we simply apply an intuitive scaling for each scale of the feature map. The scaling factor $\hat{r}_p$ is proportional to the height (scale) of the 3D feature map for each feature $f_p$ at position $p$, then we obtain scaled feature $\bar{f}_p = f_p \cdot \hat{r}_p$. Given $\bar{P}$ as the set of the 2D xy-plane positions in the voxel grid, the compressed sparse features $\bar{F}$ and their positions $\bar{P}$ are obtained as:

$$\bar{P} = \{(x_p, y_p) | p \in P\}, \quad \bar{F} = \{\sum_{p \in S_{\bar{p}}} \bar{f}_p \, | \bar{p} \in \bar{P}\}, \tag{2}$$

where $S_{\bar{p}} = \{p | x_p = x_{\bar{p}}, y_p = y_{\bar{p}}, p \in P\}$ contains voxels that are put onto the same 2D xy-plane position $\bar{p}$.

## 3.4 Network Architecture

We propose a single-stage, fully sparse neural network, designed for human pose estimation within LiDAR point clouds. The architecture is demonstrated in Figure 2. The input is a point cloud $\mathbb{R}^{N \times C}$ where $N$ is the number of points and $C$ is the number of features (*e.g.* x, y, z, intensity). We voxelize the point cloud into a sparse voxel representation. Our method consists of an input stem network and four stages with gradually decreased feature map size, where each stage reduces the spatial shape of the sparse voxel space by a factor of two. The input stem network is a simple stack of convolution layers, as shown in Appendix A.1, to extract low-level features from the voxelized point cloud. Next, four *fully sparse spatial-context blocks* are used for each subsequent stage for capturing features for accurate keypoint localization. The *sparse box-attention modules* are only applied for our last two blocks to emphasize local-region features. Note that we do not increase the number of channels for the last three stages. We then convert the resulting 3D feature maps from the last three blocks to 2D spatial-encoded BEV representations. Note that we increase the number of channels for the BEV representation to compensate for the information loss of the BEV conversion. These 2D features are further refined with 2D convolutions to aggregate spatial context. In the end, we obtain the estimated keypoints $Y_{kp} \in \mathbb{R}^{K \times 3}$ and the corresponding predicted visibilities $Y_{kp} \in \mathbb{R}^{K}$, where $K$ is the number of keypoints.

## 3.5 Relationship to Prior Works

Our *VoxelKP* fundamentally differs from the traditional LiDAR-based methods that predominantly focus on object detection or semantic segmentation tasks. Majorly, we implement the training pipeline for keypoint estimation on top of *OpenPCDet* (Team, 2020), and we use sparse convolution operators from *spconv* (Contributors, 2022). We choose *VoxelNeXt* Chen et al. (2023), a fully sparse network for 3D object detection, as the baseline architecture. To enhance the keypoint localization accuracy, our approach differs from *VoxelNeXt* from two perspectives: 1) we employ a dual-branch solution where an additional context branch is used to preserve the global spatial details, and 2) we enhance spatial awareness by projecting absolute 3D coordinates to 2D BEVs. The effectiveness of the proposed modules is supported by the ablation results in Table 2. Notably, unlike previous point-voxel blocks such as *PVCNN* (Liu et al., 2019) and *PVT* (Zhang et al., 2022), our spatial-context blocks are fully based on sparse voxels, without the need for voxelization and devoxelization within each building block. Essentially, the context branch captures per-voxel features to mitigate the loss of global context during successive convolutional blocks to ensure that each voxel retains a comprehensive understanding of its surroundings, thereby enhancing the accuracy and robustness of keypoint detection. In contrast to *PVT*'s box-attention strategy, which inefficiently handles dense tensors by repeatedly converting between dense tensors and sparse representations, we take advantage of the fully sparse architecture for a more efficient implementation.

# 4 Experiments

## 4.1 Implementation Details

***Dataset*** We use the Waymo v1.4.2 dataset (Sun et al., 2020). During the training, we merged "Pedestrian" and "Cyclist" classes together as a "Human" class. Note that there are only $8,125$ human examples with keypoint annotations whilst over 1 million bounding box annotations. We, therefore, removed the points inside those bounding boxes without keypoint annotations. Each human object is labeled with 14 3D keypoints (nose, left/right shoulders, left/right elbows, left/right wrists, left/right hips, left/right knees, and left/right ankles, head).

***Network*** The architecture of the network is composed of a stem module followed by four stages, with output channels set to 64, 128, 256, 256, and 256, respectively. Given the high resolution (*e.g.* $1504 \times 1504 \times 61$) of the voxelized point cloud input, we employ larger sparse convolution kernels (kernel size $k = 5$) for the downsampling block in both the stem module and the initial stage. For the subsequent three stages, we revert to a smaller kernel size ($k = 3$). To compensate for the information loss in the BEV projection, we increased the channels from 256 to 384.

***Training*** We use the point cloud range of $(150.4m, 150.4m, 6m)$ for the Waymo dataset and we transform them into voxel representations by a voxel size of $(0.1m, 0.1m, 0.1m)$. We directly use

the global keypoint locations without any encoding. Due to the limited number of training samples, we first apply a ground truth sampling technique (Yan et al., 2018; Chen et al., 2022) to concatenate target objects from other frames into the sampled frames. Next, we apply global augmentations on the whole point cloud, including random flips on the $x$ and $y$ axes, random scale of the range of $[0.95, 1.05]$, and random rotation ranged from $[-\pi/4, \pi/4]$. Additionally, we apply local augmentations on each annotated object, including the random scale of the range of $[0.95, 1.05]$, random rotation ranged from $[-\pi/20, \pi/20]$, random frustum dropout (Hu & Waslander, 2021) with an intensity range from $[0., 0.2]$, and random noise around the object. Our model is trained using AdamW (Loshchilov & Hutter, 2017) optimizer plus OneCycle (Smith & Topin, 2019) learning rate scheduler to mitigate overfitting (Smith, 2018). Specifically, we use a learning rate of $0.003$, weight decay of $0.01$, and $0.9$ momentum. Aside from the regular regression loss and heatmap loss, we include a skeleton regularization loss to make the model aware of the spatial relationships of keypoints. The details of the used loss functions can be found in Appendix A.2.

## 4.2 Benchmark Methods

There is a limited number of relevant research for this task. Most of the prior works utilize additional training data beyond the 3D keypoint data within the Waymo dataset. To provide a fair comparison, we need to consider approaches that use extra data and those that rely solely on Waymo ground truth separately. Zheng et al. (2022) adopted a pseudo-label generation approach to provide stronger supervision. It utilizes an internal dataset as training data and uses the Waymo dataset for evaluation. *GC-KPL* (Weng et al., 2023) pre-trains its backbone model with extra synthetic or real-world data, then fine-tunes the model with the full Waymo training set. Given the reliance on extra data in these methods, we consider the LiDAR-only version of *HUM3DIL* (Zanfir et al., 2023) as our primary competitor. *HUM3DIL* shares the exact same training data as our approach, allowing a direct comparison of techniques.

## 4.3 Results

Previous methods like *GC-KPL* use a subset of the validation data for evaluation, while we evaluate our method with the full validation set for better reproducibility. We report MPJPE on matched keypoints for our benchmark, following prior works. As shown in Table 1, we outperform the baseline *HUM3DIL* by approximately $27\%$ in MPJPE. Our approach achieves state-of-the-art results among methods trained solely on Waymo ground truth. We also surpass the approaches leveraging extra synthetic data, beating *Zheng et al.* with synthetic pseudo labels by around $18\%$ and *GC-KPL* with synthetic point clouds by about $21\%$. We achieve better performances as the SOTA *GC-KPL* approach which is pre-trained on $200,000$ real-world samples by about $12\%$. Overall, we demonstrate significant improvements over both the baseline solely using Waymo 3D keypoint data, as well as other techniques relying on extra data. Notably, our method can even achieve better results than previous multi-modal methods, *e.g.* (Zheng et al., 2022)'s multi-modal approach obtained

| Method | | Dataset | Description | MPJPE cm. |
|---|---|---|---|---|
| **With Extra Training Data** | | | | |
| Zheng *et al.* (Zheng et al., 2022) | (CVPR 22) | Internal dataset + Waymo v.? | Trained on $155,182$ objects from internal data. Generated pseudo labels from 2D image labels. | 10.80 (-18%) |
| GC-KPL (Weng et al., 2023) | (CVPR 23) | Waymo v.? | Pre-trained on synthetic data. Fine-tuned on ground truth | 11.27 (-21%) |
| | | Waymo v.? | Pre-trained on $200,000$ Waymo objects. Fine-tuned on ground truth | 10.10 (-12%) |
| **Without Extra Training Data** | | | | |
| HUM3DIL (Zanfir et al., 2023) | (CoRL 22) | Waymo v.1.3.2 | Randomly initialized | 12.21 (-27%) |
| VoxelKP | | Waymo v.1.4.2 | Randomly initialized | **8.87** |

Table 1: Benchmark results. The numbers in the table are taken from their corresponding papers aside from *HUM3DIL*, which is taken from *GC-KPL* paper. It is unclear about the exact training dataset used for *Zheng et al.* and *GC-KPL*. Waymo v1.3.2 and Waymo v1.4.2 share the same data for keypoint estimation task.

*10.32* MPJPE with both LiDAR and RGB data while we achieved *8.87* with LiDAR only. A visual demonstration is presented in Fig. 1. Please find the accompanying video in the supplementary for a visualization of the results. In addition, we report the full spectrum of the evaluation in Appendix B, including keypoint-wise MPJPE, OKS@AP, and PEM.

## 5 ABLATIONS

We demonstrate the effectiveness of each proposed component in Table 2. We use the architecture of *VoxelNext* Chen et al. (2023) as the baseline model, then gradually update the baseline model with the proposed components. We start with *VoxelNeXt* for two reasons: 1) it is one of the state-of-the-art point cloud object detection models with a fully sparse architecture design, and 2) it provides a good balance between computational costs and performance. We report the MPJPE for our ablations. The results indicate that all the individual components can contribute to improving keypoint estimation. Compared to the baseline architecture, our proposed *VoxelKP* framework improves MPJPE by 36% and PEM by 14%. Next, we further present the ablation studies to show the alternative design choices of the individual component.

| Components | | | | MPJPE | | | | | | | | PEM |
|---|---|---|---|---|---|---|---|---|---|---|---|---|
| Spatial BEV | SSK | Attention | Hybrid Feat. | head | shoulders | elbows | wrists | hips | knees | ankles | **all** | **all** |
| | | | | 0.0659 | 0.1127 | 0.1693 | 0.2020 | 0.0961 | 0.1343 | 0.1982 | 0.1394 ( 0%) | 0.1973 ( 0%) |
| ✓ | | | | 0.0737 | 0.1026 | 0.1457 | 0.2013 | 0.0878 | 0.1285 | 0.1954 | 0.1332 (+ 4%) | 0.1953 (+ 1%) |
| ✓ | ✓ | | | 0.0603 | 0.0848 | 0.1232 | 0.1715 | 0.0759 | 0.1084 | 0.1608 | 0.1118 (+20%) | 0.1889 (+ 4%) |
| ✓ | ✓ | ✓ | | **0.0558** | **0.0604** | **0.0903** | 0.1679 | **0.0620** | 0.1091 | 0.1834 | 0.1039 (+25%) | 0.1791 (+ 9%) |
| ✓ | ✓ | ✓ | ✓ | 0.0570 | 0.0669 | 0.0948 | **0.1467** | 0.0670 | **0.0820** | **0.1084** | **0.0887** (+36%) | **0.1695** (+14%) |

Table 2: Overall ablation for the effectiveness of each component. The first and last row represent the baseline *VoxelNeXt* and our proposed *VoxelKP* architecture, respectively.

### 5.1 SPATIALLY AWARE BEV

The use of the BEV representation significantly simplifies the detection problem by collapsing the 3D voxel grid into a 2D feature map. This ablation evaluates the effectiveness of the proposed spatially aware BEV module. We first evaluate the direct use of a naïve 3D representations, followed by experiments with the spatially aware BEV. The findings, as shown in Table 3, indicate that our spatially aware BEV yields superior performance. The direct deployment of the 3D representation results in severe overfitting and, therefore, low performance. In addition, we also show that increasing the number of channels during the BEV projection can effectively improve the model performance, by compensating for information loss during projection. Overall, our spatially aware BEV strikes a balance that retains spatial acuity beyond basic BEV for resolving keypoint relationships while avoiding the complexity of full 3D convolutions.

| | cp. | head | shoulders | elbows | wrists | hips | knees | ankles | all |
|---|---|---|---|---|---|---|---|---|---|
| 3D | - | 2.4620 | 2.4559 | 2.4492 | 2.4449 | 2.4394 | 2.4264 | 2.419 | 2.4422 |
| Ours | ✗ | 0.0688 | 0.0714 | 0.0982 | 0.1657 | 0.0723 | 0.1029 | 0.1595 | 0.1053 |
| Ours | ✓ | **0.0570** | **0.0669** | **0.0948** | **0.1467** | **0.0670** | **0.0820** | **0.1084** | **0.0887** |

Table 3: Ablation study for the spatially aware BEV module. *Cp.* denotes if to expand the number of channels to compensate for the information loss during the 2D projection.

### 5.2 DIFFERENT ATTENTION MECHANISM

This ablation study assesses the effectiveness of the box-attention mechanism within point cloud processing. Recent advancements, such as the stratified self-attention (Lai et al., 2022), focus on aggregating long-range contextual information, particularly beneficial for segmentation tasks. However, for keypoint estimation tasks, capturing global dependencies is less crucial. Instead, our approach utilizes local box-attention, which concentrates on adjacent local regions. The results, as presented in Table 4, demonstrate that local box-attention outperforms other methods. Interestingly, we found that the stratified attention mechanism could slightly impair performance. We suspect that the box-based approach concentrates on areas most relevant to each keypoint location, whereas long-range attention may cause the network to overlook local, dense details. As a result, the box-based

attention mechanism allows efficient modeling of local keypoint distributions, without excessive computation or over-smoothing from global aggregation.

| | head | shoulders | elbows | wrists | hips | knees | ankles | all |
|---|---|---|---|---|---|---|---|---|
| w/o | 0.0659 | 0.0956 | 0.1405 | 0.1855 | 0.0831 | 0.1077 | 0.1515 | 0.1181 |
| stratified | 0.0650 | 0.0911 | 0.1347 | 0.1995 | 0.0819 | 0.1245 | 0.1919 | 0.1266 |
| box | **0.0570** | **0.0669** | **0.0948** | **0.1467** | **0.0670** | **0.0820** | **0.1084** | **0.0887** |

Table 4: Different self-attention methods. *w/o* denotes no attention applied.

## 5.3 KEYPOINT ESTIMATION & DETECTION TRADE-OFF

Our proposed *VoxelKP* is a single-stage method that simultaneously performs both bounding box detection and keypoint estimation. Although the PEM metric accounts for penalties in both box and keypoint mismatches, to provide a clearer understanding of the performance, we report both detection metrics and keypoint estimation metrics in Table 5 under different NMS thresholds.

We reported the best MPJPE score in the main paper at an NMS threshold of 0.3, while using a threshold of 0.1 improves detection performance with a slight sacrifice in MPJPE. Compared to the *VoxelNeXt* architecture, our method achieves similar detection performance (less than 1% decrease) at an NMS threshold of 0.1, while significantly improving MPJPE performance (approximately 35% increase). At an NMS threshold of 0.3, the MPJPE performance of *VoxelKP* can be further enhanced, although it results in a more substantial loss in detection performance.

| Model | NMS Threshold | MPJPE | PEM | AP/L1 | AP/L2 | Recall@0.3 | Recall@0.5 |
|---|---|---|---|---|---|---|---|
| VoxelNeXt | 0.1 | 0.1410 | 0.2120 | **0.7147** | **0.7096** | **0.9722** | **0.9375** |
| VoxelKP | 0.1 | **0.0908** | **0.1900** | 0.7083 | 0.7049 | 0.9658 | 0.9354 |
| VoxelNeXt | 0.3 | 0.1394 | 0.1973 | **0.6665** | **0.6586** | **0.8671** | **0.8404** |
| VoxelKP | 0.3 | **0.0887** | **0.1694** | 0.6060 | 0.5998 | 0.7816 | 0.7565 |

Table 5: Keypoint estimation and object detection performances under different NMS thresholds.

## 6 CONCLUSION

In this work, we identify the challenge of learning locally dense features within a sparse environment for human keypoint estimation. We proposed a new 3D fully sparse neural network for estimating dense human poses from point clouds. We present a comprehensive solution to the intricate challenges posed by spatial information of varying densities in the context of human keypoint estimation. Our method combines several novel components including sparse selective kernel layers, box-attention layers, spatially aware multi-scale BEV fusion, and hybrid feature learning to accurately predict human body keypoints. By combining components that enhance local feature capture with those that safeguard global contextual information, our method ensures effective estimation of human keypoints. Experiments on the Waymo dataset demonstrate the advantages of our approach compared to prior art and we demonstrate improved performance compared to other approaches trained on the same data as well as other approaches trained with additional data. Through the identified challenge and the innovative framework, we pave the way for more nuanced and adaptable systems in this area.

Despite these advancements, we further identify certain areas for future exploration and improvement. As mentioned above, this work used a small volume of training data, but it could benefit from a larger-scale dataset. While we focus on single-frame point clouds, future work could leverage temporal information across sequences of LiDAR point clouds. Additionally, instead of the straightforward estimation of keypoints, future work may adopt inverse kinematics to include physical constraints on human body movement. Aside from refining estimated keypoint locations, this may especially be useful to handle real-world challenges such as occlusion within motion.

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
