# OpenReview forum: "VoxelKP: A Voxel-based Network Architecture for Human Keypoint Estimation in LiDAR Data"
_ICLR.cc/2025/Conference — ICLR 2025 Conference Withdrawn Submission_

### Official Review · Reviewer_mMQb · 2024-10-26

**Soundness:** 3
**Presentation:** 2
**Contribution:** 2
**Rating:** 5
**Confidence:** 4

**Summary:**

In this paper, authers propose a voxel-based fully sparse network, VoxelKP, for human keypoint estimation using only LiDAR point cloud data without image data. It consists of a dual-branch fully sparse spatial-context block where the spatial branch focuses on learning the local spatial correlations between keypoints within each human instance, while the context branch aims to retain the global spatial information. Then, they use a spatially aware multi-scale BEV fusion technique for the global context of each human instance. They have some fair performance comparisons on the Waymo Open Dataset with other methods and some ablation studies on different components of VoxelKP, ATTENTION MECHANISM, and the KEYPOINT ESTIMATION & DETECTION TRADE-OFF.

**Strengths:**

Simple and good idea to have a spatial branch focuses on learning the local spatial correlations between keypoints within each human instance in addition to the global spatial branch.

Good figures of the model structures and modules are very clear, making it easier for readers to understand,which improves the quality of the presentation.

Good ablation studies, good discussion about stratified self-attention on 5.2.

Have Supplementary Material to provide more details of the modules, losses, metrics, and additional results, which improves the Soundness and the quality of the presentation.

Willing to provide the code for reproducibility, which improves the Soundness.

**Weaknesses:**

The extensive use of VoxelNeXt based components makes the paper look more like an incremental work, which weakens its originality.

The lack of references to latest state-of-the-art performance results decreases the significance of this paper: MPJPE 6.72 cm from HUM3DIL (Zanfir et al., 2023), MPJPE 6.16 cm from LPFormer (Ye et al.,2023).

Some omitted expressions weaken the quality of the presentation:

In 5.1 Figure 3, please provide more details for increasing the number of channels. Although it can be inferred from the previous tables that the metrics in Table 3 are MPJPE, it would be better to specify this explicitly in the table or caption. Same as Table 4.

In Figure 1, it's better to make the wireframes thicker so that the readers can better distinguish VoxelKP from the ground truth and the baseline.

**Questions:**

Are there any ablation studies on the ground truth sampling technique and the global augmentations used during training?

Is there any performance results on the test set to ensure that there wasn't overfitting to the validation set through hyperparameter tuning?

In 5.1, how much did you increase the number of channels from and to? What limited further increases to improve performance? Better to have more analysis.

Is there any model size/runtime data to demonstrate that the performance improvement is not solely due to increasing model size/capacity?

---

### Official Review · Reviewer_bEwp · 2024-10-27

**Soundness:** 2
**Presentation:** 3
**Contribution:** 3
**Rating:** 5
**Confidence:** 4

**Summary:**

This paper proposes VoxelKP, a fully sparse network for human keypoint estimation (HPE) in LiDAR data. It is derived from a state-of-the-art detector, VoxelNeXt, and designs several techniques to fit the HPE task. By introducing SSK, Sparse Attention, and BEV fusion into HPE in LiDAR, compact representations for HPE are effectively extracted from 3D to BEV. Experiments on the large and prevailing Waymo dataset demonstrate the ability of the proposed method.

**Strengths:**

- The logic flow is clear and smooth.
- It takes successfully lots of advantages from the modern 3D object detection task in LiDAR makes it to process HPE in one-stage.
- The ablation experiment on the proposed method is detailed and convincing.

**Weaknesses:**

**Comparison on Table1 seems not fair.**

The existing two-stage methods estimate human joints in local LiDAR points. It indicates the evaluation set is based on ground truth bounding boxes or from off-the-shelf detectors. However, since the proposed method is in one stage, the result in Table1 is not under the same evaluation set. It harms the contribution in effectiveness.

The evaluation may be considered as a multi-task method: on the one hand, presenting the detection performance; on the other hand, re-implement the existing methods and compare with the bounding boxes from the proposed method. Since the main contribution is on one-stage, the "SOTA" performance against some two-stage methods seem not that important.

**Questions:**

- Note the voxel size in Waymo is set as (0.1m, 0.1m, 0.1m) while the MPJPE results is 8.87cm which is lower than the resolution of voxel. It is questionable that during fully sparse operations, the theoretical upper bound should be limited by the voxel size. Hope there is a discussion on the the sub-voxel accuracy and voxel-based limitations of one-stage method.

---

### Official Review · Reviewer_Bfm5 · 2024-11-03

**Soundness:** 3
**Presentation:** 3
**Contribution:** 2
**Rating:** 5
**Confidence:** 3

**Summary:**

This paper designs a novel fully sparse network architecture tailored for human keypoint estimation in LiDAR data. First, they introduce a dual-branch *fully sparse spatial-context block* where the spatial branch focuses on learning the local spatial correlations between keypoints within each human instance, while the context branch aims to retain the global spatial information. Second, they use a *spatially aware multi-scale BEV fusion* technique to leverage absolute 3D co-ordinates when projecting 3D voxels to a 2D grid encoding a bird’s eye view for better preservation of the global context of each human instance.

**Strengths:**

1. The spatial branch can effectively learn the local spatial correlations for human keypoints.
2. The design of the context branch is also reasonable which retain the global spatial information.
3. The BEV fusion leverage the global bird view to enhance the feature learning.

**Weaknesses:**

1. The paper writing is poor. The introduction part does not include the motivation of the proposed methods.
2. The paper lacks feature visualization to validate the effectiveness.
3. In current era, the foundation model shows the great potential to solve some perception tasks. We should validates whether the poor performance is due to the method design or the lack of the pretraining data. Could you implement your method on some pretrained model to validates the effectiveness your method?

**Questions:**

1. Please specify the motivation of your method.
2. Could you provide the feature visualization.
3. Could you validate your method based on some pretrained models?

---

### Official Review · Reviewer_3y1h · 2024-11-08

**Soundness:** 2
**Presentation:** 2
**Contribution:** 2
**Rating:** 5
**Confidence:** 2

**Summary:**

This paper presents an advancement in 3D human keypoint estimation using LiDAR data. This work addresses the inherent challenge of capturing dense local spatial details in a sparse 3D environment by introducing a fully sparse network architecture. The key contributions include a dual-branch fully sparse spatial-context block that effectively integrates local and global spatial features and a spatially aware multi-scale BEV fusion that preserves absolute 3D coordinate information during the 2D projection. Evaluated on the Waymo dataset, VoxelKP demonstrates superior performance, achieving a 27% improvement in MPJPE compared to HUM3DIL and a 12% gain over GC-KPL, even outperforming methods leveraging additional data.

**Strengths:**

This paper presents VoxelKP for 3D human keypoint estimation from LiDAR data. The paper introduces a dual-branch architecture and spatially aware BEV fusion tailored for keypoint estimation, which differentiates it from existing object detection methods.

The paper effectively demonstrates improvements over state-of-the-art methods, including those using additional data.

This work makes a significant contribution by addressing a niche yet impactful challenge in LiDAR-based keypoint estimation. The improvements demonstrated could influence related research in autonomous systems and robotics.

**Weaknesses:**

1. While the proposed method is straightforward and effective, its novelty is somewhat limited. Although VoxelKP demonstrates impressive results on the Waymo dataset, the evaluation could be broadened by incorporating more diverse datasets to better validate its robustness across different environments.

2. The empty link in the abstract is a noticeable oversight. Small errors like this can affect the professionalism of the paper and should be avoided.

3. The organization of figures and tables needs improvement for better clarity and visual coherence.

4. The experimental analysis appears somewhat limited and could benefit from deeper insights and more comprehensive comparisons, especially with the latest state-of-the-art methods.

**Questions:**

Please refer to weakness.

---

### Note · Authors · 2024-11-15

I have read and agree with the venue's withdrawal policy on behalf of myself and my co-authors.